# Injecting Frame-Event Complementary Fusion into Diffusion for Optical Flow in Challenging Scenes

**Haonan Wang[1], Hanyu Zhou[2]\*, Haoyue Liu[1], Luxin Yan[1]**
[1]National Key Lab of Multispectral Information Intelligent Processing Technology,
School of Artificial Intelligence and Automation, Huazhong University of Science and Technology
[2]School of Computing, National University of Singapore
{whn_aurora,yanluxin}@hust.edu.cn, hy.zhou@nus.edu.sg

## Abstract

Optical flow estimation has achieved promising results in conventional scenes but faces challenges in high-speed and low-light scenes, which suffer from motion blur and insufficient illumination. These conditions lead to weakened texture and amplified noise and deteriorate the appearance saturation and boundary completeness of frame cameras, which are necessary for motion feature matching. In degraded scenes, the frame camera provides dense appearance saturation but sparse boundary completeness due to its long imaging time and low dynamic range. In contrast, the event camera offers sparse appearance saturation, while its short imaging time and high dynamic range gives rise to dense boundary completeness. Traditionally, existing methods utilize feature fusion or domain adaptation to introduce event to improve boundary completeness. However, the appearance features are still deteriorated, which severely affects the mostly adopted discriminative models that learn the mapping from visual features to motion fields and generative models that generate motion fields based on given visual features. So we introduce diffusion models that learn the mapping from noising flow to clear flow, which is not affected by the deteriorated visual features. Therefore, we propose a novel optical flow estimation framework Diff-ABFlow based on diffusion models with frame-event appearance-boundary fusion. Inspired by the appearance-boundary complementarity of frame and event, we propose an Attention-Guided Appearance-Boundary Fusion module to fuse frame and event. Based on diffusion models, we propose a Multi-Condition Iterative Denoising Decoder. Our proposed method can effectively utilize the respective advantages of frame and event, and shows great robustness to degraded input. In addition, we propose a dual-modal optical flow dataset for generalization experiments. Extensive experiments have verified the superiority of our proposed method. The code is released at https://github.com/Haonan-Wang-aurora/Diff-ABFlow.

## 1 Introduction

Optical flow estimation is a visual task that models pixel-wise displacements between adjacent frames. Existing methods [11, 17, 28] focus on conventional scenes, while challenging degraded scenes such as high-speed and low-light scenes remain to be further explored. The motion blur of high-speed scenes and the insufficient illumination of low-light scenes both lead to weakened texture and amplified noise. These degradations severely deteriorate visual features and violate the photometric consistency assumption, which further brings about invalid motion feature matching.

---

\*Corresponding author.

39th Conference on Neural Information Processing Systems (NeurIPS 2025).

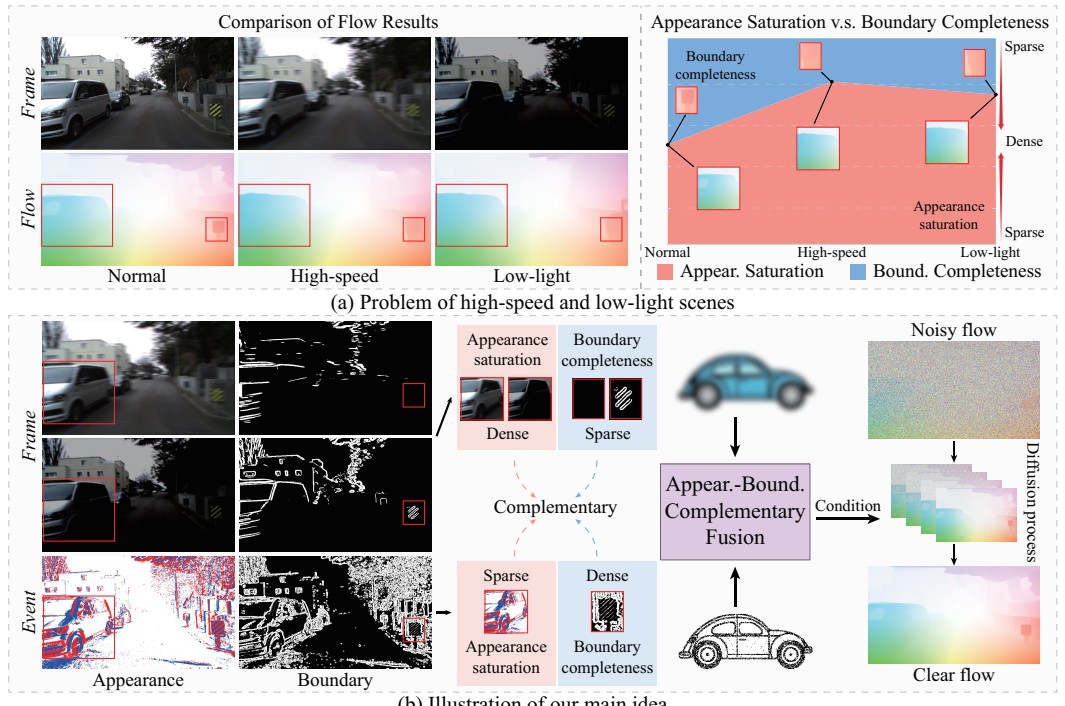

Figure 1: **Illustration of problem and idea.** Motion blur in high-speed scenes and insufficient illumination in low-light scenes reduce the boundary completeness of frame images, resulting in unclear boundary in optical flow. In this work, we explore the appearance-boundary complementarity of frame and event to guide the fusion of these two modalities. In addition, we introduce diffusion models to reconstruct the paradigm of optical flow estimation as a denoising process from noisy optical flow to clear optical flow conditioned on fused visual features.

Typically, existing methods adopt either uni-modal visual enhancement or dual-modal motion fusion to deal with high-speed and low-light conditions [8, 33, 38, 41]. The former, including deblurring and low-light enhancement improves the apparent visual effect, but the inner features remain deteriorated, contributing nothing to the photometric constancy assumption and motion feature matching. The latter utilizes event cameras to improve boundary completeness while the appearance features are still degraded and unqualified for motion feature matching. Specifically, appearance saturation refers to the abundance of appearance texture information within a visual modality. It reflects the degree of spatial variation in pixel intensity caused by fine-grained textures, shading, and color details. Boundary completeness denotes the continuity and integrity of object boundaries within a modality. It evaluates how well the modality captures clear, coherent, and complete boundary structures.

To solve these problems, we mainly explore in two aspects: sensors that can improve visual features and models that are robust to degraded input features. As shown in Fig. 1, on the one hand, we utilize the appearance-boundary complementarity of frame and event to obtain better visual features. The frame camera captures appearance with dense saturation, but due to its long imaging time and low dynamic range, the boundary captured under high-speed and low-light conditions shows sparse completeness. Thus, we introduce the event camera, a neuromorphic visual sensor [4], which offers dense boundary completeness because of its short imaging time and high dynamic range, despite its sparse appearance saturation. The appearance-boundary complementarity enables us to estimate optical flow with saturated appearance and complete boundary. On the other hand, we utilize the paradigm of diffusion models to adapt to degraded input features. Discriminative and generative models both rely on high-quality visual feature input. The former learns the mapping from visual features to motion fields, and the latter learns the process of generating motion fields based on given visual features. Both are severely affected by the degradation of visual features. Different from those two, the paradigm proposed by DDPM [9] models the denoising process from noisy data to clear data. We apply it to optical flow estimation and learn the mapping from noisy optical flow to clear optical flow, which demonstrates strong robustness to the degradation of visual features.

Based on these motivations, we propose Diff-ABFlow, a novel diffusion-based optical flow estimation framework guided by frame and event modalities. To exploit the appearance-boundary complementarity of frame and event, we propose an Attention-based Appearance-Boundary Fusion (Attention-ABF) module, which effectively combines the appearance feature of the frame and the boundary feature of the event to obtain fusion features with saturated appearance and complete boundary. Based on diffusion models, we propose a Multi-Condition Iterative Denoising Decoder (MC-IDD) as the optical flow backbone, including a Time-Visual-Motion Multi-way Cross-Attention (TVM-MCA) module and a Memory-GRU Denoising Decoder (MGDD) module. TVM-MCA integrates the fused visual feature, motion feature and temporal embedding to obtain the comprehensive feature including information from three aspects. MGDD is an optical flow inference module that combines the denoising paradigm proposed by DDIM and the iterative refinement method in optical flow estimation, which retains the generalization and robustness of diffusion models while improving efficiency. In summary, our contributions are as follows:

- We propose a novel framework, Diff-ABFlow, which leverages diffusion models with a frameevent complementary fusion strategy for accurate optical flow estimation in high-speed and low-light scenes. To the best of our knowledge, this is the first work that utilizes dual-modal data input to guide diffusion models for optical flow estimation.

- We propose the Attention-ABF module. Attention-ABF effectively utilizes the appearance-boundary complementarity of frame cameras and event cameras to obtain fusion features with high-quality appearance and boundary information.

- We propose the MC-IDD module. MC-IDD is an innovative optical flow backbone based on the DDIM paradigm and improved for optical flow estimation tasks, which combines visual features, motion features, and temporal embeddings to guide the denoising process.

- We conduct extensive experiments on both synthetic and real-world datasets to comprehensively demonstrate that our proposed Diff-ABFlow achieves state-of-the-art performance in optical flow estimation under high-speed and low-light conditions.

## 2 Related Work

**Optical Flow Estimation.** Optical flow estimation methods have developed rapidly with the advancement of deep neural networks. Earlier optical flow methods used a simple U-Net structure [3, 12]. Subsequent research gradually integrated modules such as feature pyramid and cost volume into optical flow estimation [23, 28, 35]. In addition, powerful techniques such as GRU [14, 29] and Transformer [11, 25, 34, 37] have been incorporated as the backbone for optical flow estimation. However, these frame-based methods often suffer from the motion blur in high-speed scenes and insufficient illumination in low-light scenes. In contrast, the event camera with short imaging time and high dynamic range captures high-quality visual signals especially in boundary areas. Event-based approaches [5, 7, 16, 21, 42] mainly follow the frame-based framework and reconstruct the event stream into event frame as input. In this work, we utilize the appearance-boundary complementarity of frame and event to obtain better visual features for optical flow estimation in degraded scenes.

**Degraded Scenes Optical Flow.** To deal with the motion blur of high-speed scenes and the insufficient illumination of low-light scenes, some researches directly perform visual enhancement such as deblurring and low-light enhancement. However, visual enhancement destroys the visual features and leads to invalid motion feature matching. Besides, a few methods use techniques such as feature fusion [8, 33] and domain adaptation [38, 39, 40, 41] to introduce event cameras to improve visual features. These approaches can indeed utilize event cameras to improve boundary completeness, but the appearance features provided by frame cameras are still degraded and unqualified for motion feature matching. The degradation of features severely affects discriminative models, which map from visual features to motion fields, and generative models, which generate motion fields given visual features. Therefore, we introduce diffusion models to reconstruct the optical flow estimation paradigm and reduce the impact of input visual feature degradation.

**Diffusion Model.** Diffusion models were originally proposed for image generation [9]. In contrast to previous discriminative and generative models, they model the denoising process from noisy samples to clear samples. The paradigm of diffusion models has been widely used in various fields

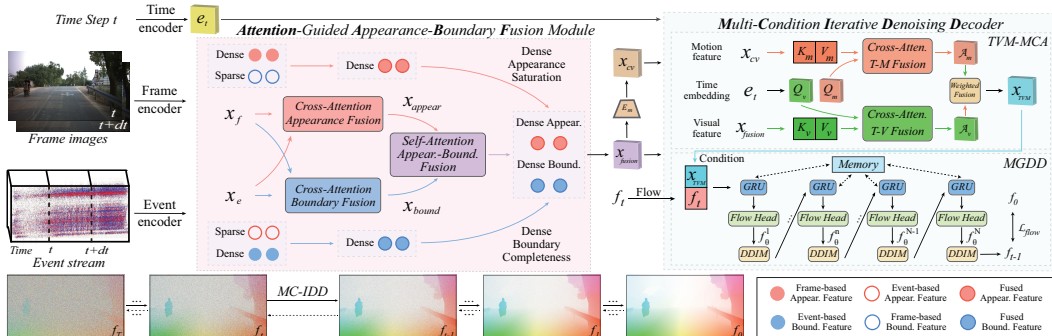

**Figure 2: Overall framework of Diff-ABFlow.** Diff-ABFlow mainly contains two parts: Attention-ABF for feature fusion and MC-IDD for denoising. In Attention-ABF, we utilize the appearance-boundary complementarity to fuse frame and event. In MC-IDD, we first integrate time embedding, visual feature and motion feature in the TVM-MCA module based on multi-way cross-attention mechanism. Then in MGDD, we input the comprehensive feature and the optical flow of the current time step into multiple GRUs with memory slots for iterative denoising. We repeatedly run MC-IDD a certain number of times on the noisy optical flow to obtain the clear optical flow.

of computer vision, such as semantic segmentation [1, 22, 30, 31, 36], depth estimation [15], trajectory prediction [13, 18]. The paradigm of these tasks has been reconstructed into the denoising process from noisy information to clear information with visual conditions. The practice in these fields has confirmed the strong robustness and generalization of diffusion models. Previous work has used diffusion models for optical flow estimation [17, 24], which reconstructed the task into a denoising process from noisy optical flow field to clear optical flow field, and achieved promising results. Therefore, we combine the appearance-boundary complementarity of frame and event, and the paradigm of diffusion models to propose a novel optical flow estimation framework that achieves state-of-the-art performance with strong generalization and robustness in degraded scenes.

## 3 Our Diff-ABFlow

### 3.1 Overall Framework

We propose a framework based on diffusion models with frame-event appearance-boundary fusion, which reformulates optical flow estimation as a denoising process from noisy flow to clear flow conditioned on frame and event. As shown in Fig. 2, the whole framework can be divided into two parts, one for feature fusion, and the other for iterative denoising based on multi-condition inputs. Based on the image pair and event stream, we extract the frame feature $x_f$ and event feature $x_e$ respectively, and input them into the Attention-ABF module to obtain the fused feature $x_{fusion}$, which is then used to construct a 4D cost volume $x_{cv}$. The time step $t$ is encoded into the time embedding $e_t$ through Sinusoidal Embedding [32] and MLP, and is then input into the TVM-MCA module together with the fused visual feature $x_{fusion}$ and motion feature $x_{cv}$ to obtain the enhanced feature $x_{TVM}$, which is finally input into the MGDD module together with the current optical flow $f_t$ for iterative denoising. The MC-IDD module is repeatedly executed to obtain a clear optical flow.

### 3.2 Attention-Guided Appearance-Boundary Fusion Module

To verify the appearance-boundary complementarity of frame and event, we design an analysis experiment to study the complementarity between frame and event at the feature level. We use the Sobel operator to extract the boundary of frame and event respectively and use K-means clustering to analyze the distribution of appearance and boundary features of frame and event. As shown in Fig. 3, the frame provides dense appearance saturation and sparse boundary completeness, while the event is the opposite. This verifies the appearance-boundary complementarity of frame and event, which motivates us to design the Attention-Guided Appearance-Boundary Fusion Module.

In the Attention-ABF module, for the input frame features $x_f$ and event features $x_e$, we first utilize appearance and boundary extractors to obtain appearance and boundary representations: $[x_{fa}, x_{fb}]$ and $[x_{ea}, x_{eb}]$. Then we obtain features with dense appearance saturation $x_{appear}$ and features with

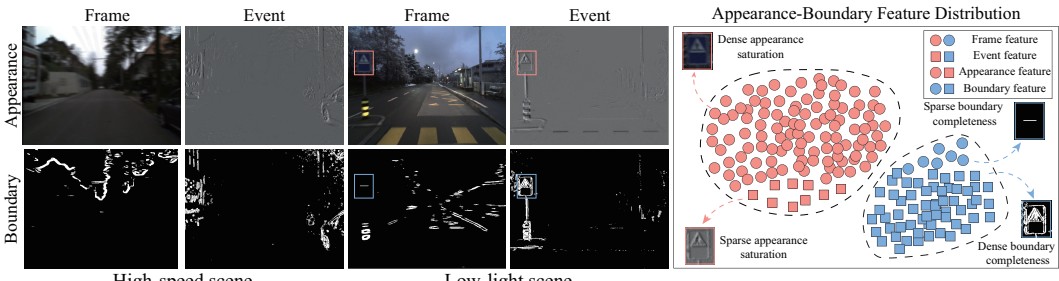

Figure 3: **Appearance-boundary feature distribution of frame and event in high-speed and low-light scenes.** We use K-means clustering to analyze the distribution of appearance and boundary features from frame and event features. The frame image has dense appearance saturation but sparse boundary completeness due to the motion blur of high-speed scenes and the insufficient illumination of low-light scenes. On the contrary, the event stream provides complete boundary in such degraded scenes while its appearance saturation is sparse. This motivates us to design a feature fusion module to fuse the two modalities utilizing the appearance-boundary complementarity.

dense boundary completeness $x_{bound}$ based on two cross-attention modules:

$$x_{appear} = \text{CAtten}(x_{fa}, x_{ea}), x_{bound} = \text{CAtten}(x_{fb}, x_{eb}). \tag{1}$$

Subsequently, we utilize the self-attention mechanism to fuse appearance and boundary information from two features: $x_{fusion} = \text{SAtten}(x_{appear}, x_{bound})$.

### 3.3 Multi-Condition Iterative Denoising Decoder

To intuitively demonstrate the superiority of diffusion models with deteriorated input features, we select a Transformer-based discriminative method and a GAN-based generative method to study the robustness to degraded inputs. Given degraded visual inputs, we utilize t-SNE to analyze the visual features and corresponding motion labels from three models. As shown in Fig. 4, both discriminative models and generative models have a certain degree of deviation when accepting degraded inputs, while the denoising process of diffusion models is almost unaffected, which motivates us to design an optical flow estimation backbone based on diffusion models, called MC-IDD.

MC-IDD utilizes the fused visual feature $x_{fusion}$, motion feature $x_{cv}$ and time embedding $e_t$ as conditions to denoise the optical flow field $\mathbf{f}_t$ at the current time step. MC-IDD includes two main parts, where TVM-MCA is used to integrate three conditions to obtain the feature $x_{TVM}$ that contains temporal, visual, and motion information, while MGDD uses the comprehensive feature $x_{TVM}$ to guide the denoising process based on GRU with memory slot.

**Time-Visual-Motion Multi-Way Cross-Attention Module.** The TVM-MCA module mainly uses two-way cross attention and gated fusion to effectively fuse time, vision, and motion features. Based on time embedding, we split it into visual query vector $Q_v$ and motion query vector $Q_m$, each of which is fused with the visual features and motion features using cross-attention, to obtain time-visual attention features $\mathcal{A}_v$ and time-motion attention features $\mathcal{A}_m$:

$$\mathcal{A}_v = Softmax\left(\frac{Q_v K_v^T}{\sqrt{d}}\right) V_v, \mathcal{A}_m = Softmax\left(\frac{Q_m K_m^T}{\sqrt{d}}\right) V_m, \tag{2}$$

where $K_v, V_v$ and $K_m, V_m$ are obtained by flattening the visual feature $x_{fusion}$ and the motion feature $x_{cv}$, and projecting them through linear layers, respectively. $d$ denotes the number of feature dimensions. Then we utilize the learnable MLP to calculate the weights $g$ of the two attention features and perform weighted fusion to obtain $x_{TVM}$:

$$x_{TVM} = g \cdot \mathcal{A}_v + (1 - g) \cdot \mathcal{A}_m, \tag{3}$$

which will be used to guide optical flow denoising later.

**Memory-GRU Denoising Decoder** Based on the paradigm of diffusion models, we first perform a forward diffusion process on the ground truth to obtain noisy optical flow, which gradually adds

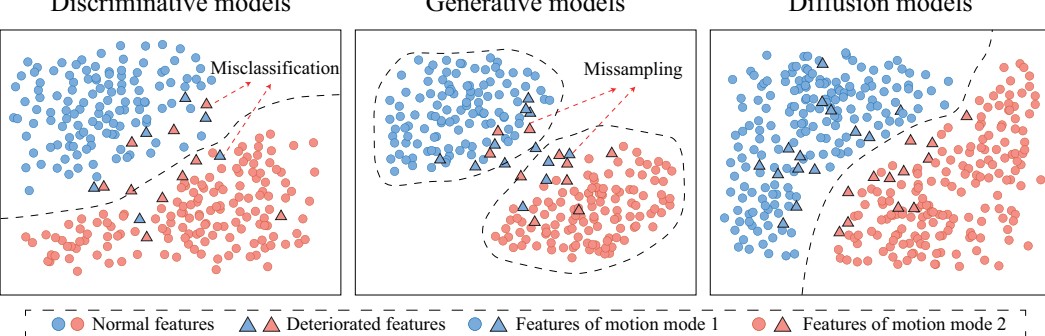

Figure 4: **t-SNE of visual features and corresponding motion labels from three different models.** Obviously, when inputting degraded features into those models, there exist misclassifications in discriminative models and missamplings in traditional generative models, while diffusion models demonstrate strong robustness to degraded inputs. This motivates us to introduce the paradigm of diffusion models and design a denoising decoding module for optical flow estimation.

Gaussian noise to the flow $T$ times using a Markovian chain. The process is formulated as:

$$q(\mathbf{f}_t \mid \mathbf{f}_0) = \mathcal{N}\left(\mathbf{f}_t \mid \sqrt{\bar{\alpha}_t}\mathbf{f}_0, (1 - \bar{\alpha}_t)\mathbf{I}\right), \quad t \in \{0, 1, \ldots, T\}, \tag{4}$$

where $\mathbf{f}_0$ indicates the ground truth of optical flow and $\mathbf{f}_t$ denotes the noisy flow. $\bar{\alpha}_t$ is defined as $\bar{\alpha}_t := \prod_{s=0}^{t} \alpha_s = \prod_{s=0}^{t}(1 - \beta_s)$, where $\beta_s$ is the pre-defined noise variance schedule, indicating the degree of Gaussian noise applied at each step.

When it comes to the reverse denoising process, our proposed MGDD module utilizes Gated Recurrent Unit (GRU) with a memory slot to iteratively denoise the optical flow $\mathbf{f}_t$. First, $x_{TVM}$ and the stored memory are jointly input into GRU as conditions and are encoded into latent features together with the optical flow. The latent features are then used to update the memory slot and input into the flow head to obtain the coarse flow prediction $\mathbf{f}_\theta^n$. The memory slot is used to store hidden features in the current iteration, which helps retain feature details at each noise level. Then we follow the DDIM [26] paradigm to calculate the denoised optical flow as Eq. 5. After $N$ iterations, the optical flow $\mathbf{f}_{t-1}$ of the next time step is obtained as:

$$\mathbf{f}_{t-1} = \sqrt{\alpha_{t-1}}\mathbf{f}_\theta^N + \sqrt{1 - \alpha_{t-1}}\,\tilde{\boldsymbol{\epsilon}}_t, \tag{5}$$

where $\tilde{\boldsymbol{\epsilon}}_t$ denotes the predicted noise at time step $t$: $\tilde{\boldsymbol{\epsilon}}_t = \frac{\mathbf{f}_t - \sqrt{\alpha_t}\,\mathbf{f}_\theta^N}{\sqrt{1-\alpha_t}}$. For the noisy optical flow $\mathbf{f}_T$, we run the MGDD module $K$ times to obtain the denoising sequence $\{\mathbf{f}_T, \mathbf{f}_{T-1}, , \mathbf{f}_{T-K}\}$.

### 3.4 Optimization

For each prediction of optical flow in the denoising sequence $\{\mathbf{f}_T, \mathbf{f}_{T-1}, , \mathbf{f}_{T-K}\}$, we introduce three loss functions to supervise the learning of the network. To supervise the optical flow prediction with the ground-truth data, we adopt an L1 loss between the predicted flow $\hat{\mathbf{f}}$ and the ground-truth flow $\mathbf{f}_0$, which is formulated as:

$$\mathcal{L}_{flow} = \|\mathbf{f}_0 - \hat{\mathbf{f}}\|, \tag{6}$$

which directly minimizes the endpoint displacement error. Then we utilize the frame to add a smoothness loss with a boundary-aware term, which encourages the predicted optical flow to be spatially smooth while preserving the boundary of optical flow:

$$\mathcal{L}_{\text{smooth}} = \sum_{x,y} \left( \left|\nabla_x\hat{\mathbf{f}}(x,y)\right| \cdot e^{-\alpha|\nabla_x I(x,y)|} + \left|\nabla_y\hat{\mathbf{f}}(x,y)\right| \cdot e^{-\alpha|\nabla_y I(x,y)|} \right) \tag{7}$$

where $I(x,y)$ denotes the first input frame and $\alpha$ is the weight of boundary-aware term. Finally we utilize event data $E_t(x,y)$ to introduce an event consistency loss, which encourages the consistency between flow and event thus improving the accuracy of optical flow in the boundary area:

$$\mathcal{L}_{event} = \sum_{x,y} \|E_t(x,y) - E_{t+1}(x + \hat{\mathbf{f}}_u(x,y), y + \hat{\mathbf{f}}_v(x,y))\| \tag{8}$$

Table 1: Quantitative results on synthetic and real datasets, where VE denotes Visual Enhancement.

| Method | Discriminative model | | | | | | | | Generative model | |
|---|---|---|---|---|---|---|---|---|---|---|
| | GMA [14] | | FF [25] | E-RAFT [7] | BFlow [8] | | ABDA-Flow [38] | | FD [17] | **Ours** |
| | w/o VE | w/.VE | - | - | w/o VE | w/.VE | - | | - | - |
| Input | Frame | | Frame | Event | Frame-event | | Frame-event | | Frame | Frame-event |
| HS-KITTI   EPE ↓ | 1.71 | 1.73 | 0.69 | 2.49 | 0.55 | 0.53 | 1.02 | | 0.62 | **0.46** |
| HS-KITTI   Fl-all ↓ | 11.44 | 12.08 | 2.18 | 16.99 | 1.90 | 1.81 | 3.27 | | 1.94 | **1.12** |
| LL-KITTI   EPE ↓ | 1.98 | 1.83 | 0.71 | 3.08 | 0.68 | 0.69 | 0.64 | | 0.67 | **0.59** |
| LL-KITTI   Fl-all ↓ | 12.36 | 11.76 | 2.85 | 19.21 | 2.54 | 2.53 | 2.46 | | 2.43 | **2.23** |
| HS-DSEC   EPE ↓ | 2.21 | 2.25 | 1.61 | 2.72 | 1.15 | 1.25 | 1.85 | | 1.17 | **1.09** |
| HS-DSEC   Fl-all ↓ | 9.65 | 10.45 | 7.32 | 13.87 | 4.13 | 4.93 | 10.43 | | 4.78 | **3.83** |
| LL-DSEC   EPE ↓ | 2.43 | 2.41 | 1.70 | 3.49 | 1.73 | 1.76 | 1.62 | | 1.69 | **1.50** |
| LL-DSEC   Fl-all ↓ | 12.78 | 12.02 | 9.65 | 18.56 | 6.48 | 6.97 | 5.69 | | 6.03 | **4.39** |

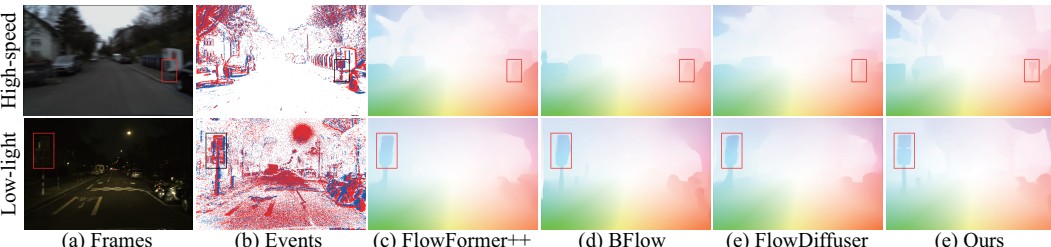

(a) Frames    (b) Events    (c) FlowFormer++    (d) BFlow    (e) FlowDiffuser    (e) Ours

Figure 5: Visualization results on real high-speed and nighttime images of HS-DSEC and LL-DSEC.

where $\hat{\mathbf{f}}_u(x, y)$ and $\hat{\mathbf{f}}_v(x, y)$ respectively denotes the horizontal and vertical components of the flow. In summary, the total loss function is formulated as:

$$\mathcal{L} = \mathcal{L}_{flow} + \lambda_{smooth} \cdot \mathcal{L}_{smooth} + \lambda_{event} \cdot \mathcal{L}_{event}, \tag{9}$$

where $\lambda_{smooth}$ and $\lambda_{event}$ are the weights for corresponding losses.

# 4 Experiments

## 4.1 Experiment Setup

**Dataset** We conducted extensive experiments on both synthetic and real datasets. The synthetic datasets, HS-KITTI and LL-KITTI, are derived from the KITTI2015 dataset [20] by applying motion blur and low-light processing, respectively, where the v2e model [10] is used to generate corresponding event streams. The real datasets include HS-DSEC, obtained by applying motion blur to the DSEC dataset [6], and LL-DSEC, which consists of nighttime segments from the original DSEC dataset. In addition, we propose a **H**igh-**S**peed **F**rame-**E**vent **F**low **D**ataset (HS-FEFD) and a **L**ow-**L**ight **F**rame-**E**vent **F**low **D**ataset (LL-FEFD), which are collected by our custom-built frame-event co-optical axis imaging device in various scenes. Note that our proposed datasets are intended for generalization evaluation and are not used for training.

**Implementation Details** For model parameters, we set the diffusion step number $T$ as 50 for forward diffusion following DDIM [26], the iterative decoding number $N$ in the MGDD module as 6, and the denoising step number $K$ as 4 for reverse denoising. During the training phase, we first pretrain the model on AutoFlow [27], FlyingChairs [3], FlyingThings [19], and MPI-Sintel [2]. Then we fine-tune it on the training sets of HS-KITTI, LL-KITTI, HS-DSEC, and LL-DSEC respectively. Finally, we conduct comparison and generalization experiments with the trained models on these datasets. All training and evaluation are performed on a single RTX 3090 GPU.

Table 2: Quantitative results on the proposed unseen HS-FEFD and LL-FEFD datasets.

| Method | | Discriminative model | | | | Generative model | |
|---|---|---|---|---|---|---|---|
| | GMA [14] | FF [25] | E-RAFT [7] | BFlow [8] | ABDA-Flow [38] | FD [17] | **Ours** |
| Input | Frame | Frame | Event | Frame-event | Frame-event | Frame | Frame-event |
| HS-FEFD EPE ↓ | 8.99 | 7.82 | 16.85 | 6.18 | 8.35 | 6.72 | **4.69** |
| HS-FEFD Fl-all ↓ | 58.11 | 57.24 | 72.35 | 45.84 | 57.96 | 49.51 | **28.77** |
| LL-FEFD EPE ↓ | 11.07 | 9.41 | 15.79 | 7.53 | 6.90 | 7.45 | **5.23** |
| LL-FEFD Fl-all ↓ | 65.82 | 59.36 | 75.89 | 55.73 | 43.31 | 52.04 | **31.49** |

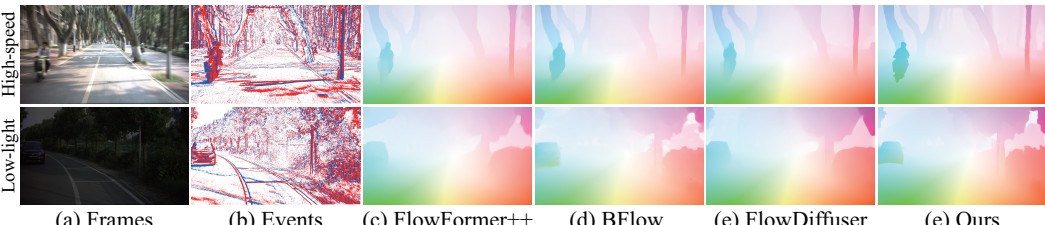

(a) Frames    (b) Events    (c) FlowFormer++    (d) BFlow    (e) FlowDiffuser    (e) Ours

Figure 6: Visualization results on the proposed unseen HS-FEFD and LL-FEFD datasets.

**Comparison Methods** We select multiple methods with different input settings and paradigms for comparison. For methods based on frame, we choose GMA [14] and FlowFormer++ (FF) [25] that use discriminative models and FlowDiffuser (FD) [17] that uses generative models. For methods based on event, we choose E-RAFT [7] and for methods based on frame-event, BFlow [8] is chosen. These methods all use the same training process as ours to ensure fairness. In addition, we add deblurring and low-light enhancement to some of these methods to test the impact of visual enhancement methods on optical flow estimation. For evaluation, we choose End-Point-Error (EPE) and the percentage of erroneous pixels (Fl-all) as metrics for quantitative evaluation.

## 4.2 Comparison Experiments

**Comparison on Synthetic Datasets.** In Table 1, we list the evaluation metrics of the proposed method and all the comparison methods on synthetic datasets. Obviously, our proposed method significantly outperforms all competing methods with different input data and different paradigms. In addition, the visual enhancement method does not significantly improve the metrics of optical flow estimation, and sometimes even makes the results worse.

**Comparison on Real Datasets.** In Table 1 and Fig. 5, we compare the proposed method with competing methods in real high-speed and nighttime scenes. First, we can conclude that the method with frame-event input performs much better than those with only frame or event input, which confirms that the complementarity of frame and event is beneficial for optical flow estimation. Second, for the methods with the same frame input, the method using diffusion models is significantly better than those using discriminative models, which verifies the excellent performance of diffusion models. Finally, the metrics and the visualization results have demonstrated the superiority of our proposed method based on diffusion models with frame-event complementarity fusion.

**Generalization for Unseen Datasets.** In Fig. 6, we compare the generalization on our proposed datasets. Under unseen real high-speed and low-light conditions, the discriminative method with frame input fails to estimate optical flow while the frame-event method performs slightly better. The generative method performs well overall, but the blurry boundary still exist. On the contrary, our proposed method works well for both appearance and boundary, reflecting strong generalization.

## 4.3 Ablation Study

**Effectiveness of Input Modality.** In Table 3, we conduct an ablation study on the input modalities. Obviously, utilizing the complementarity of frame and event data significantly improves the accuracy

Table 3: Ablation study on modalities.

| Modality | EPE ↓ | Fl-all ↓ |
|---|---|---|
| Frame | 0.58 | 1.93 |
| Event | 0.65 | 2.13 |
| **Frame+Event** | **0.46** | **1.12** |

Table 4: Discussion on fusion strategies.

| Fusion Strategy | EPE ↓ | Fl-all ↓ |
|---|---|---|
| w/ Concatenating | 0.57 | 1.84 |
| w/ Weighting | 0.55 | 1.79 |
| w/ **Attention Guided** | **0.46** | **1.12** |

Table 5: Discussion on flow estimation backbones.

| Flow Backbone | | EPE ↓ | Fl-all ↓ |
|---|---|---|---|
| Discriminative | FlowFormer | 0.59 | 1.89 |
| Generative | GAN | 0.65 | 2.97 |
| | **Diffusion Models** | **0.46** | **1.12** |

Table 6: Ablation experiments on proposed modules.

| Attention-ABF | TVM-MCA | Memory Slot | EPE ↓ | Fl-all ↓ |
|---|---|---|---|---|
| ✗ | ✗ | ✗ | 0.93 | 4.73 |
| ✗ | ✓ | ✗ | 0.65 | 2.59 |
| ✗ | ✗ | ✓ | 0.78 | 3.87 |
| ✗ | ✓ | ✓ | 0.59 | 2.09 |
| ✓ | ✗ | ✗ | 0.73 | 2.98 |
| ✓ | ✓ | ✗ | 0.54 | 1.63 |
| ✓ | ✗ | ✓ | 0.61 | 2.07 |
| ✓ | ✓ | ✓ | 0.46 | 1.12 |

Table 7: Discussion on diffusion settings.

| | Method | EPE ↓ | Fl-all ↓ | Inference Time/ms ↓ |
|---|---|---|---|---|
| Diffusion Module | U-Net | 0.63 | 2.75 | 97.5 |
| | Conditional-RDD | 0.57 | 1.74 | **63.2** |
| | **MGDD** | **0.46** | **1.12** | 64.6 |
| Denoising Steps K | 1 | 0.61 | 2.05 | **37.5** |
| | 2 | 0.52 | 1.61 | 46.4 |
| | 3 | 0.49 | 1.35 | 55.6 |
| | **4** | **0.46** | **1.12** | 64.6 |
| | 5 | 0.46 | 1.13 | 73.9 |

of the flow estimation results, achieving much better performance than using a single modality alone. This demonstrates that the two modalities provide mutually beneficial information and lead to more robust and precise flow estimation under challenging scenes.

**Influence of Proposed Modules.** In Table 6, we conduct ablation experiments on the proposed modules to reveal the effects of each module. The frame-event fusion module Attention-ABF plays the most important role in improving the results and TVM-MCA follows closely behind. The memory slot of GRU also makes a positive contribution.

## 4.4 Discussion

**How does Feature Fusion Module work?** In Fig. 7, in order to reveal the role of the feature fusion module Attention-ABF, we construct 4D cost volumes from frame features, event features, and fusion features respectively and analyze the response intensity histograms corresponding to different gradients to reflect the feature distribution in appearance and boundary areas. Moreover, we provide the flow results from the three cost volumes. On the one hand, the responses of the cost volumes from frame and event are concentrated in low-gradient and high-gradient intervals, i.e., the appearance and boundary regions, respectively, while the cost volume from the fusion features is evenly distributed in different gradient intervals. On the other hand, the flow inferred from frame has dense appearance saturation but sparse boundary completeness, and the flow inferred from event is the opposite, while the flow obtained by the fusion cost volume has dense appearance saturation and boundary completeness.This shows that our proposed fusion module effectively combines the appearance saturation and boundary completeness from frame and event.

**Impact of Feature Fusion Strategies.** In Table 4, we discuss the impact of various feature fusion strategies, including simple concatenation, weighted fusion, and our proposed attention-guided fusion. The results clearly demonstrate that the attention-guided fusion strategy significantly outperforms the other two. This superiority arises from its ability to effectively utilize the characteristics of frame and event features in appearance and boundary respectively.

**Analysis on Optical Flow Backbone.** In Table 5, we analyze the impact of different optical flow backbone architectures, including discriminative models (e.g., FlowFormer), traditional generative models (e.g., GAN-based methods), and the diffusion-based models adopted in our framework. From the results, we observe that discriminative and traditional generative models exhibit comparable performance, showing no significant differences. In contrast, diffusion models achieve substantially better accuracy and generalization in optical flow estimation.

**Choices of Diffusion Settings.** In Table 7, we conduct experiments to select the best diffusion module and denoising step. The results demonstrate that our proposed module MGDD outperforms other modules with similar inference time. In addition, the inference time increases linearly with the number of denoising steps. Thus, we set the denoising step number $K$ to 4 to achieve the best possible results without causing excessively long inference time.

Table 8: Discussion on computational efficency.

| Methods | Modules | Parameters(M) | Memory Consumption (GB) | Inference Time (ms) |
|---|---|---|---|---|
| FlowFormer++ [25] | Overall | 17.6 | 13.2 | 141.2 |
| BFlow [8] | Overall | 5.9 | 10.9 | 148.7 |
| FlowDiffuser [17] | Overall | 16.3 | 15.4 | 186.9 |
| Ours | Attention-ABF | 4.5 | 3.9 | 52.3 |
| | TVM-MCA | 3.8 | 3.5 | 46.8 |
| | MGDD | 8.9 | 8.4 | 99.4 |
| | Overall | 17.2 | 15.8 | 198.5 |

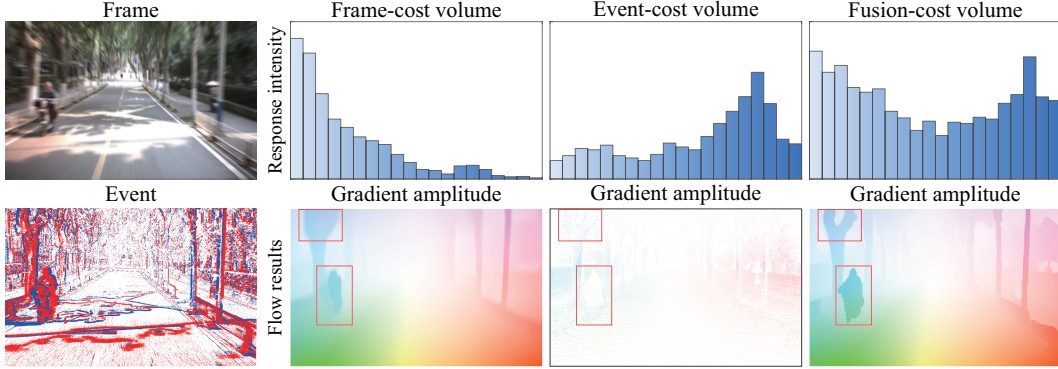

Figure 7: **Analysis on feature fusion module.** To analyze the role of the feature fusion module, we count the response intensity at different gradients of three cost volumes constructed from frame, event, and fusion features and compare the flow results from those cost volumes. The results demonstrate that the Attention-ABF module effectively utilizes the appearance-boundary complementarity of frame and event, and obtains fusion features with saturated appearance and complete boundary.

**Discussion on Computational efficiency.** For the purpose of verifying the computational efficiency of each module in our proposed method, we conduct some additional experiments on images from HS-DSEC and LL-DSEC with a resolution of $640 \times 480$ to test the number of parameters, memory consumption, and inference time of the modules and other optical flow methods, using a single RTX 3090 GPU as the inference platform. As shown in Table 8, the computational cost of each module in our proposed method remains within a reasonable range. Moreover, our approach achieves substantial performance gains with only a minor increase in computational cost.

**Limitations** Our proposed model performs well under both high-speed and low-light conditions, but fails to estimate optical flow when facing textureless planes. Neither frame nor event cameras can capture discriminative visual signals for the textureless planes since there exist no spatial brightness changes. In future work, we plan to incorporate another visual sensor, LiDAR, to perceive the distance from sensors to the planes thus obtaining the optical flow.

## 5 Conclusion

In this work, we propose a novel diffusion-based framework with event-frame appearance-boundary fusion for optical flow under both high-speed and low-light conditions. We are the first to utilize the paradigm of diffusion models with fused frame and event to solve the problem of optical flow in degraded scenes. We design the effective appearance-boundary fusion module Attention-ABF to lead the fusion of frame and event, taking advantage of their respective characteristics. In addition, we propose the innovative diffusion-based optical flow backbone MC-IDD, which aggregates information from multi-aspects including time step, visual features, and motion features, to guide the denoising process. Ours proposed method Diff-ABFlow achieves state-of-the-art performance far ahead previous methods. I believe that the multi-condition guided denoising diffusion paradigm we proposed can be used not only in the field of optical flow estimation, but also in many other fields of computer vision such as depth estimation and semantic segmentation.

## Acknowledgments and Disclosure of Funding

This work was supported in part by the National Natural Science Foundation of China under Grant U24B20139. The computation is completed in the HPC Platform of Huazhong University of Science and Technology. We also thank the reviewers for their constructive comments and suggestions.

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
