# OpenReview forum: "Injecting Frame-Event Complementary Fusion into Diffusion for Optical Flow in Challenging Scenes"
_NeurIPS.cc/2025/Conference — NeurIPS 2025 spotlight_

### Official Review · Reviewer_2Evi · 2025-06-29

**Clarity:** 2
**Significance:** 3
**Originality:** 3
**Rating:** 5
**Confidence:** 4

**Summary:**

This paper proposes an optical flow estimation method that fuses frame images and event streams, addressing the challenges posed by extreme scenarios such as high-speed motion and low-light environments. The authors argue that frame features exhibit dense appearance saturation but sparse boundary completeness, while event features are the opposite. To this end, a fusion module based on an attention mechanism is proposed to complementarily combine frame and event features. Then, based on a diffusion model, the paper introduces an iterative denoising strategy called MC-IDD to decode motion information from the fused features. Experiments on both synthetic and real datasets demonstrate the effectiveness of the approach.

**Questions:**

This paper introduces a novel idea of complementarity between frame and event data in terms of appearance and boundary information. However, the experiments are limited to selected extreme scene datasets, with no evaluation on common, non-extreme scenarios to verify generalization and applicability. Additionally, details such as hyperparameters, inference time, and module specifics are lacking. For these reasons, I give a borderline accept as my initial rating.

**Ethical Concerns:**

["NO or VERY MINOR ethics concerns only"]

**Final Justification:**

The author has addressed my concerns regarding the generalization ability of the proposed method in general scenarios. Therefore, I have raised my final rating to Accept. I also suggest that the author include inference time metrics for each method in Table 1 of the main paper.

**Limitations:**

Yes

**Quality:**

3

**Strengths And Weaknesses:**

**Strengths**:

(1)The paper clearly identifies that frame features possess dense appearance saturation but weak boundary completeness, while event features can compensate for this. The fused features contain more complete information than single-modality features.

(2)The authors propose an attention-based Frame-Event fusion module that enhances both appearance and boundary representation in the fused features.

(3)A diffusion model is used to decode motion information from the fused features via iterative denoising, and experiments show that diffusion models are more robust than discriminative and generative models.

**Weaknesses**:

(1)The authors use the Sobel operator to extract boundaries from both frames and events. However, the Sobel operator typically requires a threshold hyperparameter for non-maximum suppression, and no ablation study is provided regarding this hyperparameter.

(2)In lines 121–122, the authors mention constructing a 4D cost volume using the input frame features. However, according to RAFT [29], the cost volume is constructed by computing correlations between features of consecutive frames. Are these two methods different? If so, a comparison between them should be included.

(3)For experiments on real-world data, the authors only select extreme scene subsets (HS-DSEC and LL-DSEC) but do not specify how these sequences are split from the original DSEC dataset. To verify generalization, the method should also be evaluated on the publicly available DSEC benchmark and compared with other methods.

(4)The authors introduce two new real-world datasets: HS-FEFD and LL-FEFD, claiming them as part of their contributions. The supplementary materials mention that these datasets contain optical flow ground truth, so quantitative generalization experiments should be conducted.

(5)The inference time of Attention-ABF, MIGDD, TVM-MCA, MGDD, and the overall pipeline on KITTI and DSEC datasets should be reported and compared with other optical flow methods (e.g., BFlow [8]).

(6)In the TVM-MCA module, time embeddings from time step t are used. However, the meaning of "t" is ambiguous — is it the timestamp from the event stream and frame, or the time step in the MC-IDD denoising process? This should be clarified.

---

> ### Author Rebuttal · Authors · 2025-07-31
>
> We thank the reviewer for affirming our contributions: the pioneering exploration of the appearance-boundary complementarity between frames and events, well-designed modules, and convincing analysis experiments.
>
> Regarding the raised comments, we summarize them into six aspects: explanation of the usage of the Sobel operator, details of constructing the 4D cost volume, clarification of dataset construction and additional comparison experiments, additional quantitative generalization experiments, comparison of computational efficiency, and explanation of some notations.
>
> ### Q1: Explanation of the usage of the Sobel operator.
>
> **A:** Thank you for your detailed review comments. Actually we did not perform ablation experiments on the threshold of the Sobel operator because the Sobel operator was only used to obtain boundary maps in the analysis experiment in Fig. 3. In practice, we set the threshold of non-maximum suppression in the Sobel operator to 50 to get a good boundary map visualization result. If necessary, we will provide boundary map visualizations for different thresholds in the final version of the paper to validate our choices.
>
> ### Q2: Details of constructing the 4D cost volume.
>
> **A:** The method we use to construct the 4D cost volume is similar to RAFT [1]. The difference between the two lies in the input. RAFT uses two consecutive frames as input to calculate correlations. Our input $x_{fusion}$ is the visual feature obtained by fusing the features of two consecutive frames and the features of the event stream at the corresponding timestamps through Attention-ABF module. Then, we calculate correlations between the fusion visual features $x_{fusion}$ at two timestamps to obtain the 4D cost volume as the motion feature.
>
> ### Q3: Clarification of dataset construction and additional experiments on the original DSEC [2] dataset.
>
> **A:** Actually we mentioned the relevant explanation in lines 196-197 of the paper. The HS-DSEC is obtained by applying motion blur to daytime segments from the original DSEC dataset, and the LL-DSEC is composed of nighttime segments from the original DSEC dataset. Moreover, in order to verify the generalization of our proposed method, we conducted some additional experiments to compare the performance of our proposed method with other comparison methods in optical flow estimation on the original DSEC dataset. Specifically, we randomly separated 3888 images from the original DSEC dataset as the training set to train our proposed method and all comparison methods, and the remaining 432 images were used as the test set to test the optical flow estimation performance metrics of each method. As shown in the table below, our proposed method achieves the state-of-the-art performance on the original DSEC dataset for optical flow estimation.
>
> |  Scenes  | Metric | GMA [3] | FlowFormer++ [4] | E-RAFT [5] | BFlow [6] | FlowDiffuser [7] | Ours |
> | :------: | :----: | :-----: | :--------------: | :--------: | :-------: | :--------------: | :--: |
> | DSEC [2] |  EPE   |  1.43   |       1.25       |    2.54    |   1.12    |       1.07       | 0.95 |
> | DSEC [2] | F1-all |  5.89   |       5.38       |    9.84    |   4.26    |       3.54       | 2.73 |
>
> ### Q4: Additional quantitative generalization experiments on HS-FEFD and LL-FEFD.
>
> **A:** Following the reviewer's comment, we conduct additional quantitative generalization experiments on HS-FEFD and LL-FEFD for all methods. As shown in the table below, owing to the advantages of event camera to capture clear and complete boundaries in high-speed and low-light scenes, and the strong generalization and robustness of diffusion models, our proposed method achieves the best performance.
>
> | Scenes  | Metric | GMA [3] | FlowFormer++ [4] | E-RAFT [5] | BFlow [6] | FlowDiffuser [7] | Ours  |
> | :-----: | :----: | :-----: | :--------------: | :--------: | :-------: | :--------------: | :---: |
> | HS-FEFD |  EPE   |  8.99   |       7.82       |   16.85    |   6.18    |       6.72       | 4.69  |
> | HS-FEFD | F1-all |  58.11  |      57.24       |   72.35    |   45.84   |      49.51       | 28.77 |
> | LL-FEFD |  EPE   |  11.07  |       9.41       |   15.79    |   7.53    |       7.45       | 5.23  |
> | LL-FEFD | F1-all |  65.82  |      59.36       |   75.89    |   55.73   |      52.04       | 31.49 |
>
> ### Q5: Comparison of inference time between the modules in our proposed method and other optical flow methods.
>
> **A:** In order to comprehensively test the computational efficiency of the proposed method, we conducted some additional experiments on images from KITTI with a resolution of 1242×375 and DSEC with a resolution of 640×480 to test the number of parameters, memory consumption, and inference time of the proposed method and the comparison methods, using a single RTX 3090 GPU as the inference platform.
>
> As shown in the table below, we can observe the following:
>
> - The computational efficiency of each module in our proposed method is within a reasonable range.
> - Resulting from the additional introduction of event modality and the nature of iterative denoising of diffusion models, although the overall metrics of our method are slightly higher than those of existing methods, it achieves a significant improvement in performance.
>
> |     Methods      |    Modules    | Parameters (M) | Memory Consumption on KITTI/DSEC (GB) | Inference Time on KITTI/DSEC (ms) | EPE on KITTI/DSEC |
> | :--------------: | :-----------: | :------------: | :-----------------------------------: | :-------------------------------: | :---------------: |
> | FlowFormer++ [4] |    Overall    |      17.6      |               17.3/13.2               |            179.5/141.2            |     0.73/1.67     |
> |    BFlow [6]     |    Overall    |      5.9       |               15.5/10.9               |            187.6/148.7            |     0.67/1.46     |
> | FlowDiffuser [7] |    Overall    |      16.3      |               19.4/15.4               |            231.3/186.9            |     0.65/1.44     |
> |       Ours       | Attention-ABF |      4.5       |                5.2/3.9                |             68.1/52.3             |         -         |
> |       Ours       |    TVM-MCA    |      3.8       |                4.8/3.5                |             53.2/46.8             |         -         |
> |       Ours       |     MGDD      |      8.9       |               10.9/8.4                |            126.9/99.4             |         -         |
> |       Ours       |    Overall    |      17.2      |               20.9/15.8               |            248.2/198.5            |     0.49/1.23     |
>
> ### Q6: Explanation of the meaning of $t$ in the TVM-MCA module.
>
> **A:** $t$ denotes the denoising time step, which acts as a controller, enabling the denoising module to perform adaptive denoising behavior at different denoising time steps. This is achieved by encoding the denoising time step $t$ into the time embedding $e_t$ and fusing it with motion/visual features $x_{cv}/x_{fusion}$ via a cross-attention mechanism, which is subsequently served as a conditioning signal for the denoising process.
>
> ### Reference
>
> [1] Zachary Teed, et al. Raft: Recurrent all-pairs field transforms for optical flow. *ECCV* 2020.
>
> [2] Mathias Gehrig, et al. Dsec: A stereo event camera dataset for driving scenarios. *RAL* 2021.
>
> [3] Shihao Jiang, et al. Learning to estimate hidden motions with global motion aggregation. *ICCV* 2021.
>
> [4] Xiaoyu Shi, et al. Flowformer++: Masked cost volume autoencoding for pretraining optical flow estimation. *CVPR* 2023.
>
> [5] Mathias Gehrig, et al. E-raft: Dense optical flow from event cameras. *3DV* 2021.
>
> [6] Mathias Gehrig, et al. Dense continuous-time optical flow from event cameras. *TPAMI* 2024.
>
> [7] Ao Luo, et al. Flowdiffuser: Advancing optical flow estimation with diffusion models. *CVPR* 2024.

---

### Official Review · Reviewer_NzT1 · 2025-06-29

**Clarity:** 3
**Significance:** 3
**Originality:** 2
**Rating:** 5
**Confidence:** 3

**Summary:**

The paper proposes Diff-ABFlow, a novel diffusion-based framework for optical flow estimation in high-speed and low-light scenes using complementary fusion of frame and event data. It introduces two key modules: Attention-ABF, which enhances feature fusion by leveraging appearance boundary complementarity, and MC-IDD, an improved DDIM-based backbone tailored for optical flow tasks. Extensive experiments demonstrate that Diff-ABFlow achieves state-of-the-art performance across multiple datasets.

**Questions:**

1. It is recommended that the authors provide optical flow estimation results under specific weather conditions, which may help to better understand the robustness of the proposed model.
2. Why do some comparison methods in Table 1 show performance degradation after visual enhancement? Please provide an explanation.

**Ethical Concerns:**

["NO or VERY MINOR ethics concerns only"]

**Final Justification:**

The authors' rebuttal resolved my concerns. Therefore, I have raised my score

**Limitations:**

yes

**Paper Formatting Concerns:**

No formatting issues

**Quality:**

3

**Strengths And Weaknesses:**

Strengths:
1. This paper utilizes the appearance-boundary complementarity of frame and event data to obtain better visual features for optical flow estimation in degraded scenes, and the motivation is clear.
2. The experiments in the paper demonstrate that the proposed method performs well on both synthetic and real-world data, forming an effective validation loop in the experimental setup.

Weaknesses:
1. This work shares a similar motivation with ABDA-Flow (ICLR '24), as both use RGB and event data for complementary appearance and boundary information, leading to overlap in novelty.
2. In lines 198–199 of the paper, the authors claim to have proposed the HS-FEFD and LL-FEFD datasets. However, the visualized images in Figure 6 are identical to those in Figure 6 of ComST-Flow (CVPR '25). The authors are requested to clarify this.
3. It is recommended that the authors include a comparison with ABDA-Flow (ICLR '24) in Table 1 and add its visualizations in Figure 5.

---

> ### Author Rebuttal · Authors · 2025-07-31
>
> We thank the reviewer for affirming our contributions: the pioneering exploration of the appearance-boundary complementarity between frames and events, clear motivation, and abundant experiments.
>
> Regarding the raised comments, we summarize them into five aspects: difference from ABDA-Flow, clarification of the visualized images, comparison experiments with ABDA-Flow, generalization under other challenging conditions, and explanation of the performance degradation of some methods after visual enhancement.
>
> ### Q1: Differences between the proposed method and ABDA-Flow [1].
>
> **A:** It is emphasised that our method is different from ABDA-Flow in three typical aspects: task, motivation and framework.
>
> - Regarding the task, ABDA-Flow focuses on spatial degradation in low-light scenes caused by the global exposure of frame cameras, which leads to texture loss in dark regions. In contrast, our method additionally addresses temporal degradation in high-speed scenes, where fixed exposure times cause motion blur in fast-moving objects.
>
>
> - Regarding the motivation, while ABDA-Flow leverages the high dynamic range of event cameras to mitigate spatial degradation, it does not resolve temporal degradation. Our work explores the appearance-boundary complementarity between frames and events to handle both spatial degradation in low-light scenes and temporal degradation in high-speed scenes.
>
>
> - Regarding the framework, ABDA-Flow employs a unidirectional transfer from the event to the frame domain to enhance spatial quality, leveraging the high dynamic range of events. In contrast, our proposed method exploits the appearance-boundary complementarity between frames and events and introduces a bidirectional complementary fusion strategy, effectively addressing both types of degradation.
>
> Overall, the task with temporal and spatial degradation in high-speed and low-light scenes, the motivation with frame-event appearance-boundary complementarity , and the framework with a bidirectional complementary fusion strategy and diffusion-based module jointly constitute a significant innovation of our method.
>
> ### Q2: Clarification of the visualized images in Fig. 6.
>
> **A:** Thanks to the reviewer for pointing out this issue. As described in the supplementary materials, the HS-FEFD and LL-FEFD datasets used for generalization testing in Fig. 6 of our paper were collected by our frame-event dual-mode system and subsequently processed. Regarding the appearance of similar images in related studies, it's possible that the authors used the same data source as ours to construct their own datasets. We can confirm that the datasets used in our generalization experiments were collected by ourselves in real-world scenes. Further details about these datasets will be made public upon acceptance of the paper.
>
> ### Q3: Additional comparison with ABDA-Flow [1].
>
> **A:** We did not compare with ABDA-Flow before because the author did not open source the code. During the rebuttal period, we contacted the author by email to obtain the source code of ABDA-Flow and conducted additional comparative experiments. As shown in the table below, we can observe the following:
>
> - ABDA-Flow performs well in low-light scenes but fails in high-speed scenes since the method is designed to focus on nighttime conditions.
> - Our proposed method achieves state-of-the-art performance in both high-speed and low-light scenes.
>
> |  Scenes  | Metric |  GMA  | FlowFormer++ | E-RAFT | BFlow | ABDA-Flow [1] | FlowDiffuser | Ours |
> | :------: | :----: | :---: | :----------: | :----: | :---: | :-----------: | :----------: | :--: |
> | HS-KITTI |  EPE   | 1.71  |     0.69     |  2.49  | 0.55  |     1.02      |     0.62     | 0.46 |
> | HS-KITTI | F1-all | 11.44 |     2.18     | 16.99  | 1.90  |     3.27      |     1.94     | 1.12 |
> | LL-KITTI |  EPE   | 1.98  |     0.71     |  3.08  | 0.68  |     0.64      |     0.67     | 0.59 |
> | LL-KITTI | F1-all | 12.36 |     2.85     | 19.21  | 2.54  |     2.46      |     2.43     | 2.23 |
> | HS-DSEC  |  EPE   | 2.21  |     1.61     |  2.72  | 1.15  |     1.85      |     1.17     | 1.09 |
> | HS-DSEC  | F1-all | 9.65  |     7.32     | 13.87  | 4.13  |     10.43     |     4.78     | 3.83 |
> | LL-DSEC  |  EPE   | 2.43  |     1.70     |  3.49  | 1.73  |     1.62      |     1.69     | 1.50 |
> | LL-DSEC  | F1-all | 12.78 |     9.65     | 18.56  | 6.48  |     5.69      |     6.03     | 4.39 |
>
> ### Q4: Generalization of the proposed method for other challenging conditions.
>
> **A:** It is worth noting that our method aims to address the optical flow estimation problem in high-speed, low-light scenes. Our main idea is to exploit the  the complementarity between frames and events in appearance saturation and boundary completeness to obtain better optical flow in high-speed and low-light scenes.
>
> Following the reviewer's comment, we conduct additional experiments on rainy, foggy, large-displacement, and overexposed scenes. For rainy and foggy scenes, we use the Weather-KITTI2015 (Rain-KITTI, Fog-KITTI) and Weather-GOF (Rain-GOF, Fog-GOF) introduced by Zhou et al. [2], with synthetic events generated using the v2e model [3]. For large displacements, we constructed the LD-DSEC dataset by sampling one frame every three frames and re-segmenting events from the original DSEC, using the method of Ce Liu et al. [4] to generate optical flow ground truth. For overexposed scenes, we built the OE-DSEC dataset by selecting frames with intense car headlights or streetlights from DSEC. Each dataset includes 800 randomly selected images for generalization testing. We additionally include CH$^2$DA-Flow [2], an unsupervised method for optical flow in adverse weather, as a comparison baseline.
>
> As shown in the table below, we can observe the following:
>
> - From the overall perspective, our proposed method performs well in large displacement and overexposed scenes, but perform not so well in the rainy and foggy scenes.
> - In large-displacement and overexposed scenes, our method significantly outperforms others. The fixed temporal resolution and exposure time of frame cameras cause temporal discontinuity and texture loss in bright regions, while event cameras, with their high temporal resolution and dynamic range, capture continuous motion and preserve scene details under such challenging conditions.
> - In rainy and foggy scenes, our proposed method performs poorly, since raindrops and fog not only blur the frame image but also introduce a large number of noise points in the event stream, making it difficult for our method to obtain good visual features.
>
> In summary, our proposed method performs better under conditions where we can obtain good visual features through the appearance-boundary complementarity between frames and events. However, under conditions where both frame cameras and event cameras are limited, we will explore solutions such as domain adaptation in the future.
>
> |   Scenes   | Metric |  GMA  | FlowFormer++ | E-RAFT | BFlow | CH$^2$DA-Flow [2] | FlowDiffuser | Ours  |
> | :--------: | :----: | :---: | :----------: | :----: | :---: | :---------------: | :----------: | :---: |
> | Rain-KITTI |  EPE   | 8.02  |     7.54     | 18.73  | 6.27  |       5.78        |     4.97     | 4.23  |
> | Rain-KITTI | F1-all | 46.31 |    41.35     | 75.72  | 37.55 |       33.59       |    26.72     | 24.92 |
> |  Rain-GOF  |  EPE   | 6.03  |     4.97     | 14.57  | 4.22  |       3.54        |     3.28     | 3.19  |
> |  Rain-GOF  | F1-all | 35.73 |    31.62     | 57.35  | 21.74 |       22.75       |    18.94     | 17.36 |
> | Fog-KITTI  |  EPE   | 7.59  |     6.75     | 17.54  | 6.19  |       5.98        |     5.72     | 4.89  |
> | Fog-KITTI  | F1-all | 43.62 |    38.91     | 67.91  | 37.29 |       38.94       |    32.47     | 28.74 |
> |  Fog-GOF   |  EPE   | 7.14  |     5.47     | 15.12  | 4.68  |       3.87        |     3.79     | 3.65  |
> |  Fog-GOF   | F1-all | 44.29 |    36.84     | 59.68  | 26.91 |       24.52       |    24.77     | 22.83 |
> |  LD-DSEC   |  EPE   | 5.45  |     4.23     |  8.62  | 3.52  |       4.18        |     3.79     | 2.67  |
> |  LD-DSEC   | F1-all | 34.82 |    30.52     | 47.34  | 21.64 |       28.96       |    24.83     | 12.45 |
> |  OE-DSEC   |  EPE   | 5.68  |     3.91     |  9.43  | 3.04  |       3.96        |     3.25     | 2.38  |
> |  OE-DSEC   | F1-all | 37.04 |    28.79     | 55.86  | 17.28 |       27.43       |    18.42     | 10.94 |
>
> ### Q5: Explanation of the performance degradation of some methods after visual enhancement.
>
> **A:** Actually, we mentioned the relevant explanation in lines 34-35 of the paper. The visual enhancement such as motion deblurring and low-light enhancement indeed improves the apparent visual effect of frame images. However, such improvement may have a negative effect on optical flow estimation since visual enhancement may destroy the pixel matching relationship between adjacent frames, resulting in incorrect optical flow results.
>
> ### Reference
>
> [1] Hanyu Zhou, et al. Exploring the Common Appearance-Boundary Adaptation for Nighttime Optical Flow. *ICLR* 2024.
>
> [2] Hanyu Zhou, et al. Adverse Weather Optical Flow: Cumulative Homogeneous-Heterogeneous Adaptation. *TPAMI* 2024.
>
> [3] Yuhuang Hu, et al. v2e: From video frames to realistic dvs events. *CVPR* 2021.
>
> [4] Ce Liu, et al. Human-Assisted Motion Annotation. *CVPR* 2008.

---

> > ### Comment · Reviewer_NzT1 · 2025-08-05
> >
> > Thanks for the authors' rebuttal, which resolved my concerns. The paper's motivation is clear, using event cameras to address low-light, low-texture challenges in optical flow estimation, with sound logic. New generalization experiments across weather conditions in the rebuttal confirm its effectiveness. As the first diffusion-based multimodal optical flow work with high-quality datasets (HS-FEFD, LL-FEFD),  I’d increase my score and recommend acceptance. Publicly releasing the dataset would enhance its value to the community.

---

> ### Author Response · Authors · 2025-08-05
> **Thank You for the Comments**
>
> Thank the reviewer for your positive feedback. We are glad that you increased your score and recommended acceptance. We will update all the comments in the revised version.

---

### Official Review · Reviewer_qEsh · 2025-07-02

**Clarity:** 3
**Significance:** 2
**Originality:** 3
**Rating:** 5
**Confidence:** 3

**Summary:**

The paper presents a framework called Diff-ABFlow for optical flow estimation in high-speed and low-light scenes. The authors propose using diffusion models combined with frame-event appearance-boundary fusion to address the challenges posed by motion blur and insufficient illumination. The framework leverages the complementary strengths of frame cameras (dense appearance saturation) and event cameras (dense boundary completeness) to improve optical flow estimation. Key components include the Attention-Guided Appearance-Boundary Fusion module and the Multi-Condition Iterative Denoising Decoder . Extensive experiments demonstrate the superiority of the proposed method over existing approaches.

**Questions:**

1.	Paper states: ‘Kv, Vv and Km, Vm are obtained by flattening the visual feature xf usion and the motion feature xcv, respectively.’ Does this imply that no additional projection layer is applied—so that K_v is actually identical to V_v, and likewise K_m to V_m, with the different symbols used only for notation?
2.	How does the proposed method perform in other challenging conditions, such as foggy or rainy environments? It would be valuable to test the framework in a wider range of scenarios.

**Ethical Concerns:**

["NO or VERY MINOR ethics concerns only"]

**Final Justification:**

The paper presents a framework called Diff-ABFlow for optical flow estimation in high-speed and low-light scenes. Most of my questions are addressed in the rebuttal. Therefore, I maintain my score.

**Limitations:**

The authors have discussed the limitations of their work, particularly the difficulty in estimating optical flow for textureless planes. They propose incorporating LiDAR sensors in future work to address this issue. However, the paper does not explicitly discuss potential negative societal impacts.

**Paper Formatting Concerns:**

There are no formatting concerns in this paper.

**Quality:**

3

**Strengths And Weaknesses:**

Strengths:
+ The Attention-ABF module combines the strengths of frame and event cameras, leading to improved visual features. Dual modal features are used to guide diffusion models for optical flow estimation.
+ Multi-Condition Iterative Denoising Decoder are used to progressively denoise the optical flow. Detailed ablation studies are conducted to demonstrate the contribution of each modules.
+ The authors conduct extensive experiments on multiple datasets, including synthetic and real-world scenarios, to validate their method.

Weaknesses:
- The proposed framework is complex, involving multiple modules and iterative processes, which may pose challenges for implementation and computational efficiency.
- The method is primarily tested on high-speed and low-light scenes, and its generalizability to other challenging conditions is not thoroughly explored.

---

> ### Author Rebuttal · Authors · 2025-07-31
>
> We thank the reviewer for affirming our contributions: the pioneering exploration of utilizing dual modal features to guide diffusion models for optical flow estimation, detailed ablation studies, and extensive experiments.
>
> Regarding the raised comments, we summarize them into three aspects: implementation and computational efficiency of the method, generalization under other challenging conditions, and clarification of some notations.
>
> ### Q1: Implementation and computational efficiency of our proposed method.
>
> **A:** The framework seems complex, but is simply modularized into two components: Attention-ABF and MCIDD. The former performs appearance-boundary complementary fusion of frames and events through cross-attention mechanism and self-attention mechanism, and the latter adopts cross-attention mechanism to aggregate information from multiple conditions, including time steps, visual features, and motion features, which is subsequently used to guide the optical flow denoising process.
>
> In order to compare the computational efficiency of these modules with different optical flow methods, we conduct some additional experiments on images from HS-DSEC and LL-DSEC with a resolution of 640×480 to test the number of parameters, memory consumption, and inference time of the methods, using a single RTX 3090 GPU as the inference platform.
>
> As shown in the table below, we can observe the following:
>
> - The computational efficiency of each module in our proposed method is within a reasonable range.
> - Resulting from the additional introduction of event modality and the nature of iterative denoising of diffusion models, although the overall metrics of our method are slightly higher than those of existing methods, it achieves a significant improvement in performance.
>
> |   Methods    |    Modules    | Parameters (M) | Memory Consumption (GB) | Inference Time (ms) | EPE  |
> | :----------: | :-----------: | :------------: | :---------------------: | :-----------------: | :--: |
> | FlowFormer++ |    Overall    |      17.6      |          13.2           |        141.2        | 1.67 |
> |    BFlow     |    Overall    |      5.9       |          10.9           |        148.7        | 1.46 |
> | FlowDiffuser |    Overall    |      16.3      |          15.4           |        186.9        | 1.44 |
> |     Ours     | Attention-ABF |      4.5       |           3.9           |        52.3         |  -   |
> |     Ours     |    TVM-MCA    |      3.8       |           3.5           |        46.8         |  -   |
> |     Ours     |     MGDD      |      8.9       |           8.4           |        99.4         |  -   |
> |     Ours     |    Overall    |      17.2      |          15.8           |        198.5        | 1.23 |
>
>
>
> ### Q2: Generalization of the proposed method for other challenging conditions.
>
> **A:** It is mentioned that our method is designed to solve the problem for optical flow estimation in high-speed and low-light scenes. Our main idea is to exploit the  the complementarity between frames and events in appearance saturation and boundary completeness to obtain better optical flow in high-speed and low-light scenes.
>
> Following the reviewer's comment, we conduct additional experiments on rainy, foggy, large-displacement, and overexposed scenes. For rainy and foggy scenes, we use the Weather-KITTI2015 (Rain-KITTI, Fog-KITTI) and Weather-GOF (Rain-GOF, Fog-GOF) introduced by Zhou et al. [1], with synthetic events generated using the v2e model [2]. For large displacements, we constructed the LD-DSEC dataset by sampling one frame every three frames and re-segmenting events from the original DSEC, using the method of Ce Liu et al. [3] to generate optical flow ground truth. For overexposed scenes, we built the OE-DSEC dataset by selecting frames with intense car headlights or streetlights from DSEC. Each dataset includes 800 randomly selected images for generalization testing. We additionally include CH$^2$DA-Flow [1], an unsupervised method for optical flow in adverse weather, as a comparison baseline.
>
> As shown in the table below, we can observe the following:
>
> - From the overall perspective, our proposed method performs well in large displacement and overexposed scenes, but perform not so well in the rainy and foggy scenes.
> - In large-displacement and overexposed scenes, our method significantly outperforms others. The fixed temporal resolution and exposure time of frame cameras cause temporal discontinuity and texture loss in bright regions, while event cameras, with their high temporal resolution and dynamic range, capture continuous motion and preserve scene details under such challenging conditions.
> - In rainy and foggy scenes, our proposed method performs poorly, since raindrops and fog not only blur the frame image but also introduce a large number of noise points in the event stream, making it difficult for our method to obtain good visual features.
>
> In summary, our proposed method performs better under conditions where we can obtain good visual features through the appearance-boundary complementarity between frames and events. However, under conditions where both frame cameras and event cameras are limited, we will explore solutions such as domain adaptation in the future.
>
> |   Scenes   | Metric | GMA [4] | FlowFormer++ [5] | E-RAFT [6] | BFlow [7] | CH$^2$DA-Flow [1] | FlowDiffuser [8] | Ours  |
> | :--------: | :----: | :-----: | :--------------: | :--------: | :-------: | :---------------: | :--------------: | :---: |
> | Rain-KITTI |  EPE   |  8.02   |       7.54       |   18.73    |   6.27    |       5.78        |       4.97       | 4.23  |
> | Rain-KITTI | F1-all |  46.31  |      41.35       |   75.72    |   37.55   |       33.59       |      26.72       | 24.92 |
> |  Rain-GOF  |  EPE   |  6.03   |       4.97       |   14.57    |   4.22    |       3.54        |       3.28       | 3.19  |
> |  Rain-GOF  | F1-all |  35.73  |      31.62       |   57.35    |   21.74   |       22.75       |      18.94       | 17.36 |
> | Fog-KITTI  |  EPE   |  7.59   |       6.75       |   17.54    |   6.19    |       5.98        |       5.72       | 4.89  |
> | Fog-KITTI  | F1-all |  43.62  |      38.91       |   67.91    |   37.29   |       38.94       |      32.47       | 28.74 |
> |  Fog-GOF   |  EPE   |  7.14   |       5.47       |   15.12    |   4.68    |       3.87        |       3.79       | 3.65  |
> |  Fog-GOF   | F1-all |  44.29  |      36.84       |   59.68    |   26.91   |       24.52       |      24.77       | 22.83 |
> |  LD-DSEC   |  EPE   |  5.45   |       4.23       |    8.62    |   3.52    |       4.18        |       3.79       | 2.67  |
> |  LD-DSEC   | F1-all |  34.82  |      30.52       |   47.34    |   21.64   |       28.96       |      24.83       | 12.45 |
> |  OE-DSEC   |  EPE   |  5.68   |       3.91       |    9.43    |   3.04    |       3.96        |       3.25       | 2.38  |
> |  OE-DSEC   | F1-all |  37.04  |      28.79       |   55.86    |   17.28   |       27.43       |      18.42       | 10.94 |
>
> ### Q3: Clarification of $K_v, V_v$ and $K_m, V_m$.
>
> **A:** I apologize for not clarifying how we obtain $K_v, V_v$ and $K_m, V_m$. Specifically, we first flatten the visual feature $x_{fusion}$ and motion feature $x_{cv}$ into tokens, and consequently project them into $K_v, V_v$ and $K_m, V_m$ with the same dimension as $Q_v$ and $Q_m$ by learnable linear layers, respectively. Therefore, $K_v$ is actually not identical to $V_v$, and likewise $K_m$ to $V_m$.
>
> ### Reference
>
> [1] Hanyu Zhou, et al. Adverse Weather Optical Flow: Cumulative Homogeneous-Heterogeneous Adaptation. *TPAMI* 2024.
>
> [2] Yuhuang Hu, et al. v2e: From video frames to realistic dvs events. *CVPR* 2021.
>
> [3] Ce Liu, et al. Human-Assisted Motion Annotation. *CVPR* 2008.
>
> [4] Shihao Jiang, et al. Learning to estimate hidden motions with global motion aggregation. *ICCV* 2021.
>
> [5] Xiaoyu Shi, et al. Flowformer++: Masked cost volume autoencoding for pretraining optical flow estimation. *CVPR* 2023.
>
> [6] Mathias Gehrig, et al. E-raft: Dense optical flow from event cameras. *3DV* 2021.
>
> [7] Mathias Gehrig, et al. Dense continuous-time optical flow from event cameras. *TPAMI* 2024.
>
> [8] Ao Luo, et al. Flowdiffuser: Advancing optical flow estimation with diffusion models. *CVPR* 2024.

---

> > ### Comment · Reviewer_qEsh · 2025-08-05
> >
> > Thank you to the authors for their thorough response to my concerns. Most of the issues have been well addressed with detailed quantitative results, particularly regarding complexity and performance under various conditions. Although the proposed method introduces a slight increase in complexity, this is acceptable given the substantial performance improvements. Regarding generalization, new results under different settings have been provided and discussed appropriately. I will maintain my original score.

---

> > > ### Author Response · Authors · 2025-08-06
> > >
> > > Thank you for recognizing our work and recommending acceptance. We will update all the comments in the revised version.

---

### Official Review · Reviewer_dJAt · 2025-07-03

**Clarity:** 2
**Significance:** 3
**Originality:** 3
**Rating:** 5
**Confidence:** 4

**Summary:**

This paper proposes a diffusion-based method for event-guided optical flow estimation. Its main contribution is a novel feature fusion strategy that employs triple-branch cross- and self-attention to condition the DDIM process, along with innovative designs in the diffusion denoiser incorporating GRUs, memory slot, and other elements. Experimental results demonstrate the effectiveness of the proposed method.

**Questions:**

1. What is meant by “appearance saturation”? Does it refer to the abundance of appearance textures within a modality? If so, what do “dense” or “sparse” appearance saturation specifically mean — do they indicate an abundance or lack of appearance textures? Although these terms do not obscure the paper’s overall motivation (i.e., using events for boundary information and frames for appearance features, which are complementary for optical flow estimation), they can be confusing upon first reading.

2. Why do the authors apply cross-attention between the time embedding and the two modality features? The time embedding primarily encodes the diffusion step index and does not appear to share semantic information with the modality features. Is the intention for the time embedding to act as a bridge between the two modalities? Clarifying this design choice would be helpful.

**Ethical Concerns:**

["NO or VERY MINOR ethics concerns only"]

**Final Justification:**

I have read the authors' rebuttal, which addresses most of my concerns. Therefore, I have raised my score.

**Limitations:**

yes

**Paper Formatting Concerns:**

No Formatting Concerns.

**Quality:**

3

**Strengths And Weaknesses:**

Strengths
  1. The application of diffusion models to event-based optical flow has not been widely explored.
  2. The experimental results appear promising.
  3. The graphical illustrations are clear and helpful.

Weaknesses
  1. There are other works that employ diffusion models for optical flow estimation, such as FlowDiffuser (CVPR 2024), which also uses a GRU-based denoiser. What distinguishes this work from FlowDiffuser? Simply adding an event modality with a cross-attention fusion mechanism may not constitute a significant innovation.
  2. The clarity of the paper could be improved. For instance, terms like sparse/dense appearance saturation are confusing. Saturation is typically used to describe colors, and its usage here is unclear and distracting.

---

> ### Author Rebuttal · Authors · 2025-07-31
>
> We thank the reviewer for affirming our contributions: the pioneering exploration of diffusion models for event-based optical flow, promising experimental results, clear and helpful graphical illustrations.
>
> Regarding the raised comments, we summarize them into three aspects: the difference from FlowDiffuser, clarification of some concepts, and methodological details.
>
> ### Q1:Differences between the proposed method and FlowDiffuser.
>
> **A:** It is emphasized that our method is different from FlowDiffuser in three typical aspects: task, motivation and framework.
>
> (1) Regarding the task, FlowDiffuser is designed for optical flow estimation in conventional scenes. In contrast, our proposed method explores the temporal and spatial degradation of frame images caused by motion blur in high-speed scenes and texture loss in low-light scenes, which in turn leads to inaccurate optical flow estimation.
>
> (2) Regarding the motivation, FlowDiffuser introduces diffusion models with promising performance for optical flow estimation to obtain a more generalizable and robust model. In addition, we introduce event modality and exploit the appearance-boundary complementarity between frames and events to solve the temporal and spatial degradation of frame images in high-speed and low-light scenes.
>
> (3) Regarding the framework, FlowDiffuser simply designs an optical flow iterative denoising decoder based on diffusion models. Differently, we propose Attention-ABF and MCIDD. The former adopts cross-attention module to focus on the appearance information of the frame modality and the boundary information of the event modality, respectively, and subsequently fuse them through the self-attention module to obtain better visual features. The latter integrates information from time steps, visual features, and motion features to guide the denoising process.
>
> Overall, the task with optical flow estimation in high-speed and low-light scenes, the motivation with frame-event appearance-boundary complementarity, and the framework with diffusion models guided by fused frame and event features jointly constitute a significant innovation of our method.
>
> ### Q2: Clarification of “appearance saturation”.
>
> **A:** We apologize for causing your misunderstanding on the concept of appearance saturation.
>
> In this paper, appearance saturation refers to the abundance of appearance texture information within a visual modality. It reflects the degree of spatial variation in pixel intensity caused by fine-grained textures, shading, and color details. Dense appearance saturation refers to the abundance of fine-grained textures, shading, and color details, while sparse appearance saturation is the opposite. Modalities with dense appearance saturation, such as frame images, preserve abundant structural and photometric cues, which are helpful for establishing correlations between two consecutive frames and further estimating optical flow. In practice, appearance saturation can be quantified by measuring the variance of local gradient magnitudes, emphasizing the presence of detailed visual patterns:
> $$
> S_{appear} = \mathrm{Var} \left( \sqrt{ \left( \frac{\partial I}{\partial x} \right)^2 + \left( \frac{\partial I}{\partial y} \right)^2 } \right),
> $$
> where $ I $ denotes frame image or event frame constructed from original event stream.
>
> In addition, we also provide the definition of boundary completeness. Boundary completeness denotes the continuity and integrity of object boundaries within a modality. It evaluates how well the modality captures clear, coherent, and complete boundary structures. Dense boundary completeness means that the visual modality has clear, continuous, and complete boundaries, while the boundaries of a visual modality with sparse boundary completeness may be blurred, interrupted, or even missing. For modalities with dense boundary completeness, such as events, the captured boundaries are clear and tightly connected in space, which plays an important role in obtaining optical flow with clear and complete boundaries. To quantitatively calculate boundary completeness, we first obtain the boundary map through the Sobel operator, and then calculate the ratio of the number of pixels of connected boundaries in the boundary map to all boundary pixels as the boundary completeness:
> $$
> S_{bound} = \frac{ \sum_{c=1}^{C} L_c }{ \sum_{i=1}^{N} L_i}，
> $$
> where $ L_c,L_i $ denote the pixel length of connected boundaries and all boundaries, respectively. $C,N$ are the number of connected boundaries and the number of all boundaries, respectively.
>
> Due to motion blur in high-speed scenes and insufficient illumination in low-light scenes, frame cameras with long imaging time and low dynamic range have dense appearance saturation but sparse boundary completeness, while event cameras with short imaging time and high dynamic range have dense boundary completeness but sparse appearance saturation. Therefore, we exploit the complementarity between frames and events in appearance saturation and boundary completeness to design a fusion module to obtain better visual features.
>
> ### Q3: Explanation of the design of using cross-attention mechanism between time embedding and motion/visual features.
>
> **A:** It is worth noting that we actually apply the cross-attention mechanism between time embedding $e_t$ and motion/visual features $x_{cv}/x_{fusion}$,
>
> rather than between time embedding and two modality features. Rather than serving as a bridge between the two modalities, the time embedding acts as a controller, enabling the denoising module to perform adaptive denoising behavior at different time steps. Specifically, in the initial stage with more noise, the module should remove as much noise as possible, while in the later stage with less noise, it should consider preserving the optical flow details while removing noise. Based on this motivation, we use the cross-attention mechanism between time embedding and motion/visual features in order to enable the model to learn to adaptively extract features according to the denoising time step. The features are subsequently fused to guide the denoising process, thereby realizing the regulation of the denoising behavior by the diffusion time step.

---

> > ### Comment · Reviewer_dJAt · 2025-08-03
> > **Thank You for the Rebuttal**
> >
> > I thank the authors for the detailed response. My concerns are mostly addressed. However, I still believe the diffusion part of this work is similar to FlowDiffuser. As the authors noted, MCIDD integrates information from time steps, visual features, and motion features to guide the denoising process, just as the Conditional-RDD in FlowDiffuser does (which also incorporates motion features, visual/context features, and timestep information).
> >
> > Perhaps the memory slot design is a distinguishing factor. In FlowDiffuser, the decoder GRU’s hidden features (analogous to the memory slot in this work) are initialized from visual features. During training, since the diffusion timestep is randomly sampled, after initialization, the hidden features only apply within a single diffusion step. For inference, the hidden state is retained across all DDIM diffusion steps.
> >
> > How is the memory slot in this work initialized? Does it serve as a global variable, regardless of timestep, training, or inference? If so, how do you handle the issue of random timesteps during training? For instance, a memory slot updated for t=1000 in the current data batch may not be appropriate for training samples with t=1 sampled in the next batch.
> >
> > Another minor issue is in Fig. 2, where multiple DDIM blocks are shown within each GRU iteration. Is this depiction accurate? It appears that DDIM steps are being interleaved with GRU iterations, which seems inconsistent with the standard setup. My understanding is that there should be a single DDIM step following multiple GRU iterations. If this interpretation is incorrect, I would appreciate clarification.
> >
> > Overall, I acknowledge the effort in integrating modality fusion and attention design into a solid event- and diffusion-based optical flow framework. In light of this, I have raised my score and recommend the paper for acceptance.

---

> > > ### Author Response · Authors · 2025-08-04
> > > **Thank You for the Comments**
> > >
> > > The reviewer’s comments mainly focus on three aspects: similarity between our proposed method and FlowDiffuser, details of the memory slot, and clarification of the DDIM modules.
> > >
> > > (1) Regarding the similarity between our proposed method and FlowDiffuser, we acknowledge that the diffusion module in our proposed method is similar in general idea to that in FlowDiffuser, as both follow the DDIM paradigm and apply it to optical flow estimation, a task that requires information from visual features and motion features. However, our method further explores conditioning signals to better handle challenging scenes. Moreover, we slightly improve the implementation details of the optical flow denoising module, which will be discussed in the next two responses.
> > >
> > > (2) Regarding the details of the initialization of the memory slot, it is distinguished from the hidden features of FlowDiffuser and is initialized from the comprehensive feature $x_{TVM}$. During training, it is updated to store the features denoised by the GRU module, which is used in the subsequent GRU module. After a MGDD module comprising a certain number of GRU modules, we resample the denoising step and reinitialize the memory slot with $x_{TVM}$. During inferencing, it is retained across the entire denoising process. In summary, the memory slot is designed to store and propagate valuable features throughout the denoising process.
> > >
> > > (3) Regarding the clarification of the DDIM modules, while the standard setup uses a single DDIM step after multiple GRU iterations, we apply a DDIM step after each GRU iteration so that we can perform multiple denoising iterations within a single MGDD module. We find that this strategy denoises the optical flow more effectively, with only a small increase in computational cost.

---

> > > > ### Comment · Reviewer_dJAt · 2025-08-05
> > > > **Thanks**
> > > >
> > > > I appreciate the additional clarification, which I think is reasonable. I have no further questions.

---

> > > > > ### Author Response · Authors · 2025-08-05
> > > > > **Thanks**
> > > > >
> > > > > Thank the reviewer for your recognizing. We are glad that you raised your score and recommended our paper for acceptance. We will update all the comments in the revised version.

---

### Decision · Program_Chairs · 2025-09-17

**Decision:**

Accept (spotlight)

**Comment:**

This paper introduces Diff-ABFlow, a novel framework for optical flow in challenging scenes that effectively fuses frame and event data within a diffusion model. The approach is technically sound, well-motivated by the complementary nature of the two modalities, and demonstrates strong performance. The authors' engagement during the rebuttal period was exemplary, providing a remarkable number of substantial new experiments to address every major reviewer concern.

While the initial submission faced valid questions regarding its novelty compared to recent work and the generalization of its results, the rebuttal effectively transformed the paper. The authors conducted new, direct comparisons against key baselines and comprehensively tested their method on a wider variety of adverse conditions, thoroughly resolving all initial points of contention.

This exceptional effort was reflected in the final reviewer stances, creating a unanimous positive consensus and causing all reviewers to raise their scores in support of the paper. Therefore, because the paper now stands as a solid and comprehensively validated contribution, my recommendation is Accept.